# Decrease in Carabid Beetles in Grasslands of Northwestern China: Further Evidence of Insect Biodiversity Loss

**DOI:** 10.3390/insects13010035

**Published:** 2021-12-29

**Authors:** Xueqin Liu, Xinpu Wang, Ming Bai, Josh Jenkins Shaw

**Affiliations:** 1School of Agriculture, Ningxia University, Yinchuan 750021, China; liuxueqin@nxu.edu.cn; 2Key Laboratory of Zoological Systematics and Evolution, Institute of Zoology, Chinese Academy of Sciences, Beijing 100101, China; joshjenkins@btinternet.com

**Keywords:** biodiversity decline, carabid beetles, environmental change, insect conservation, long-term monitoring, steppe, species richness trends

## Abstract

**Simple Summary:**

Long-term studies on insect diversity trends are rare, especially in steppe ecosystems. To gain insights into carabid beetle diversity trends in steppe grasslands, we analyzed data on carabids from a trapping study that ran for 12 years in the grasslands of northwestern China. We found that species abundance and richness declined over time. The results of this study suggest that precipitation may play a role in changing species dynamics. This study emphasizes the urgent need to protect carabid communities in steppe ecosystems.

**Abstract:**

Ground-dwelling beetles are important functional components in nutrient-poor grasslands of middle temperate steppe ecosystems in China. Here, we assessed the changes in ground beetle (Coleoptera: Carabidae) communities in the grasslands of northwestern China over 12 years to improve the management and conservation of beetles all over the world. The Generalized Additive Model (GAM) was applied to estimate the changes in carabid beetle communities in two regions: a desert steppe (Yanchi region), and a typical steppe and meadow steppe (Guyuan region). During the 12-year investigation, a total of 34 species were captured. We found that species abundance and richness per survey declined by 0.2 and 11.2%, respectively. Precipitation was the main factor affecting the distribution of carabid beetles. A distinct decline in carabid beetle species in the Yanchi region indicated that they may be threatened by less precipitation and loss of habitat, which could be due to climate change. Overall, species richness was stable in the Guyuan region. It is necessary to estimate and monitor the changes in carabid beetle communities in a temperate steppe of northern China and to protect them. Extensive desertification seriously threatens the distribution of carabid beetles. Future research should develop methods to protect carabid beetle communities in temperate steppes in China.

## 1. Introduction

It is well known that the loss of insects has a serious impact on ecosystem function [1], as insects play a central role in ecosystems, in terms of pollination and nutrient cycling [2,3,4,5] and has a significant economic impact [6,7,8]. For example, 75% of the world’s major food crops show increased fruit or seed sets with animal pollination [9]. It has been estimated that ecosystem services provided by insects are worth 57 billion USD annually in the USA [10]. Therefore, preserving insect abundance and diversity should be a priority for insect conservation.

Dramatic declines in insect abundance and richness have caused widespread concerns among ecologists and insect scientists. Declines in the number of insects have been confirmed worldwide [11,12,13,14]. These declines can potentially threaten ecological ecosystems [15]. It is still unclear whether the declines are widespread. For example, a reliable analysis of changes in the distribution of insect species is still absent. Given the current rate of global environmental change, quantifying changes in species abundance is critical in assessing ecosystem impacts. However, long-term monitoring data related to insect population trends are scant, especially in grassland ecosystems, because many studies mainly focus on forest and agricultural ecosystems [16,17,18,19]. Therefore, preserving insect abundance and richness is critical in the development of conservation strategies in grassland ecosystems.

Many studies also have shown substantial declines in some insect groups, suggesting that insect biodiversity loss is more severe than other estimates [20,21,22]. For example, insect abundance decreased by 45% after evaluating two-thirds of the taxa [2], and annual grasshopper abundance declined by 2.1–2.7% per year in 16 time series from a Kansas prairie [23]. In addition, the decline in specific insects with different target taxa and habitats has been reported [20]. Insects play an important role in ecosystems, and their decline is likely to cause an imbalance in the natural ecosystem [24]. Grassland is one of the natural and semi-natural environment types with high species abundance. The effect of grasslands on insect community structure and population changes has become an important issue in the study of species diversity over the world [25,26]. However, how the insects are reduced in steppe ecosystems remains unresolved. Long-term species-level statistical data are scant, and information on species trends for many key insect groups critical to ecosystems is lacking [27]. Carabid beetles (Coleoptera: Carabidae), one of the largest families of insects, are well-fitted for such investigations [9]. They can be used as indicators of changes in habitats and environments and play an important role in assessing environmental impact and conservation [28,29,30]. They react more promptly than most long-living animals or plants due to their relatively short life cycles [7,31]. Carabid beetles live in a wide variety of biotopes and play a crucial role in functional ecology because many species prey on pest species and plant seeds [32]. For example, in an agricultural ecosystem, carabid beetles are important because they are natural enemies to many pests [33]. A decline in the abundance of carabid beetles causes concerns over food safety. Some believe that it is impossible to maintain high productivity with decreasing chemical inputs [34]. Consequently, it is essential to explore carabid beetles for long-term studies, which also can enhance the understanding of diversity loss.

In all studies, how species respond to environmental change is clarified, but complex biotic interactions are the most important in understanding diversity loss [35,36]. For example, climate and habitats can change species dynamics [12], and habitat loss is most likely to cause insect extinctions as community structure, the physical environment, and community stability change [37]. Furthermore, theoretical methods to model large-scale changes in biodiversity indicate that species response is almost equal in time and space [38]. Indeed, the spatiotemporal dynamics of species richness is important for knowing how environmental variations impact biodiversity [39,40]. All of these endeavors increase our understanding of the influence of potential driving factors over habitats and regions [41,42,43]. Therefore, models used to estimate trends in geographic changes in many species and their interaction with habitats and regions are needed to increase our understanding of biodiversity loss.

In China, steppes, which account for 80% of grassland vegetation, have high biodiversity and conservation value. However, steppes are being degraded into barren land and deserts probably due to climate change and human activities [44]. Such habitat degradation has a dramatic impact on steppe biodiversity. Since 2008, carabid beetles along with many climatic and physiological variables have been monitored within a range of grassland across northwest China [45,46,47]. Thus, previous data can be used to learn the mechanisms that control biodiversity. Although growing evidence of carabids in northwest China showed that their range of distributions have declined, most of which are believed to be caused by climate change [42], the shortage of simultaneously collected and quantitative data has impeded precise predictions of species trends. Here, we selected sites with representative steppes and repeated beetle surveys from 2008 to 2019. Those surveys involved all available natural habitats and regions in the survey area, including desert steppes, typical steppes, and meadow steppes. To reveal the effects of climate change and increased eutrophication due to land-use intensification, we analyzed soil and vegetation parameters in parallel with our beetle monitoring. We used those data to test hypotheses: (i) beetle species have gradually disappeared from the Yanchi region (desert steppe) because of increasing temperature and decreasing precipitation, and (ii) beetle communities of the Guyuan region (typical and meadow steppes) have remained relatively stable. These results may promote assessments of population changes and may help understand how the biodiversity of important functional groups changes over time in regions and habitats of northwestern China.

## 2. Materials and Methods

### 2.1. Study Area

From 2008 to 2019, 135 samples sites were surveyed over ~6000 km^2^ in two regions of Ningxia Province in northwestern China: Yanchi region (YC) and Guyuan region (GY). These regions are temperate steppes between 35° N and 37° N. The species richness of carabid beetles in these two regions are the highest compared with other insects (e.g., Scarabaeidae and Tenebrionidae) (Figure 1).

(1) The Yanchi region is dominated by desert steppes, with a dominance of sierozem soils, semi-arid continental monsoonal climate, and low humidity (annual mean temperature 8.3 °C; 200 mm of precipitation annually) [48]. The vegetation in this study area is characterized by *Agropyron mongolicum*, *Artemisia desertorum*, *Lespedez adavurica*, and *Artemisia blepharolepis*.

(2) The Guyuan region is dominated by meadow steppes and typical steppes, with a dominance of black thorn and brown soils, semi-arid continental monsoonal climate, and moderate humidity (annual mean temperature 7 °C; 400 mm of precipitation annually) [48]. Some representative vegetation include *Stipa bungeana*, *Artemisia frigida*, *Potentilla acaulis*, and *Stipa grandis*.

### 2.2. Beetle Samples

There were 31 sample sites in the Yanchi region and 104 sample sites in the Guyuan region. The three steppe types at these sites were sampled in 2008, 2009, 2013, 2017, 2018, and 2019. From 2008 to 2009, 6 of the 135 samples sites were resampled between six and eight times, yielding 220 sample events. In 2013, 5 of the 135 samples were resampled once each, yielding 125 sample events. In 2017, 90 of the 135 samples were resampled (see Table A1), with a total of 1800 sample events. In 2018 and 2019, 34 of the 135 samples were resampled, with a total of 1360 sample events.

We used pitfall traps (400 mL capacity and 7.5 cm diameter) placed with the top of the trap flush with the soil surface to catch carabid beetles. All sites, the number of sampling rounds, and their dates were synchronized. The surveys were conducted continuously from May to September of every sampling year since, in this period, they are active. At each sample site, there were five pitfall traps at 5 m intervals, which were separated by 200–1400 m from each other to minimize spatial autocorrelation. Traps inducing liquid were placed with the top of the trap being flush with the soil surface and were collected three days later. During this time, beetles were stored in 75% ethanol and counted and identified in the laboratory [49]. Here, we pooled the traps per site in a 20 × 20 m sample and received 79 effective sample sites. We accounted for the number of beetles once a month and took the average of five measurements for analysis. Carabid beetles have significant single, annual activity peaks, and each site includes at least five pitfall traps that can be used to estimate the abundance of localized carabid beetles. The beetles were collected in a standard way through traps and were classified by the Chinese Academy of Science.

### 2.3. Soil and Environment Parameters

The soil and vegetation parameters were determined on each survey date. The soil temperature was recorded at a depth of 0–10 cm, and the humidity and temperature were measured using a moisture analyzer (YH-SWP-100, Jiangsu VICTOR Instrument Mete Company, Taizhou, China). The vegetation parameters were measured within a 1 m^2^ quadrat frame. The climatic data were extracted from WorldClim (https://www.worldclim.org, monthly weather data for 1960–2018). 

### 2.4. Data Analysis

Changes in beetle distribution with time were quantified using two measures: species richness and abundance. The variations in species richness (i.e., number of species) and abundance are evaluated per trap among 2008 to 2019. The sampling size as a covariable was included in the models. The abundance was calculated per survey to include a co-variate as an explanatory variable in the models [50]. Species abundance and richness per trap and Shannon–Wiener diversity were calculated with iNEXT R package [51].

We used GAM to model beetle richness and abundance with the mgcv package in R (version 4.0.3) [52]. These models are extensions of the Generalized Linear Model (GLM) framework [53], and one “best model” is constructed [54]. It is important to avoid using collinear explanatory variables in GAM. GAM was used to assess nonlinear trends in the effect response curve. The goodness factor of the competing functions was measured using an F-ratio with a 5% significance level. For model assessment, the evidence ratio, AIC, and minimized generalized cross validation (GCV) score were applied [55]. Latitude, elevation, soil temperature, vegetation cover, plant species, annual mean temperature, and annual mean precipitation were included in the model. As beetle data are usually distributed skewedly, its corresponding probability density functions are in lognormal or gamma distributions, which exclude data points at zero. We used the Tweedie distribution, which is better suited for data with zero values.

Finally, we used Canonical Correspondence Analysis (CCA) to analyze the explanatory variables. The soil moisture, soil temperature, vegetation coverage, plant diversity, mean annual temperature, mean annual precipitation, altitude, and year were regarded as environmental variables that were selected by a stepwise model selection using the backstep function.

## 3. Results

### 3.1. Beetle Abundance and Species Richness

In total, 16,830 specimens belonging to 34 different species from all sampled sites since 2008 were recorded (see Table A1). Table 1 shows the descriptive statistics of the species richness and abundance of carabid beetles in the Yanchi and Guyuan regions. The species richness in the Yanchi and Guyuan regions ranged from 1 to 17 and 6 to 16 per sample, respectively. The abundance ranged from 2 to 523 and 30 to 268 per sample, respectively. The Guyuan region had the highest average species richness (9.32) and abundance (92.95), followed by the Yanchi region (5.26, 62.91) (Table 1 and Table A2).

At all sites combined, the year had an obvious negative influence on species richness and abundance (Table 2). The species richness and abundance were substantially lower in later years than in earlier ones. The abundance decreased over time but revealed an overall decrease, with the strongest declines being from 2008 to 2013 (Figure 2). The species richness and abundance per survey decreased over time by approximately 0.2% (Figure 2a) and 11.2% (Figure 2b), respectively.

### 3.2. Measure Diversity

Species richness, mean abundance, and Shannon diversity were significantly lower in the Yanchi region than in the Guyuan region (Figure 3). Overall, there were no significant temporal trends in Shannon diversity, but individual plots showed signs of decline in abundance.

Our model showed the major explanatory descriptors of variation within the pooled data from the two regions (Figure 4). We recorded 29 species from Guyuan region, 27 species from Yanchi region, and 22 species from both the Guyuan and Yanchi regions (Table A2). The year had a significant influence on species richness and abundance. Species richness also significantly influenced by the soil moisture and precipitation (Table A3, Figure 4a). Abundance decreased yearly and increased with latitude and precipitation. It was also significantly influenced by plant diversity and soil temperature (Table A3, Figure 4b).

### 3.3. Pattern of Spatial Distribution

The pattern of species distribution differed between the YC and GY region: the abundance decreased in YC and increased in GY (Figure 5a); the pooled species richness remained relatively stable from 2008 to 2019 (Figure 5b).

The CCA indicated that all parameters of the variables had a significant effect (Figure 6). CCA axes 1–4 cumulatively explained 85.50% (31.9, 57.75, 71.64, and 85.5) of the total constrained variations (Table A4). The main gradient was defined by altitude, soil temperature, and precipitation. Year is another main independent parameter, with increasing temperature being the most closely related one.

## 4. Discussion

Consistent with recent long-term studies of insects, we observed a decline in species richness and abundance, which were in accordance with other studies on insect taxa, such as moths, butterflies, and bees [24]. These studies showed that insect diversity is declining. Concerns are exacerbated for carabid beetles because they play an important role within an ecosystem [32]. Although the decline was lower than with other insects, the species richness can modulate pest control as a predator [56]. In summary, the insect diversity being threatened is supported by these data.

For all surveys, year had a significant negative effect on abundance and species richness. Given that carabid abundance and richness have mostly downward trends, they can be attributed as being the dominant driver of change. Species losses and decreases in total abundances cannot be attributed to natural succession. Although grasslands have high species diversity, it has become a threatened habitat. Our analyses revealed that the abundance of carabid beetles is declining when compared with other groups of insects, supporting the evidence that this group of species is severely threatened [23,57,58,59]. Concerns about carabids were aggravated because they were frequently used to indicate environmental changes [25,32]. All of those supported the hypothesis that there was a significant reduction in the abundance of carabid species over time and that those changes also led to a loss of biodiversity. Therefore, at the regional scale, those rates are lethal for carabids.

Our results can contribute to understanding biodiversity loss and its management at the regional level [60]. The easiest way to consider the decline is to attribute it to a dominant driver of change. However, there were important differences in the responses between richness and abundance of carabids, suggesting a more complex interaction between species drivers and the environment. We found wide-scale changes in the carabids in the Yanchi region because of regional differences caused by climate change. For example, temperature increase was disproportionately greater in the desert steppe, with concomitant reductions in precipitation. The decrease in soil moisture and increase in soil temperature adversely affect the breeding of carabids, resulting in the abundance of the species declining most in the Yanchi region. Conversely, increasing temperature and abundant precipitation provided more suitable conditions for the breeding of carabids in the Guyuan region. Species from the Guyuan region seemed to be less influenced by the destruction of habitats. This means that the GY region provides more favorable microclimatic conditions for beetles, such as befitting temperature and humidity, in contrast with the YC region. One important reason is probably the increased precipitation and temperature in these grasslands in the second half of the last century [61]. Carabids are intolerant to freezing [62]; therefore, low temperature and precipitation impede their breeding.

With the desertification of land, the grasslands of northwestern China have seriously degenerated in the study area, causing a serious decline in biodiversity [25]. For all surveyed regions, the year had a significant negative effect on species richness and abundance. The trend of beetle decline was in line with other studies [6,63]. The sharp decline in beetles has been recorded in a similar situation in European countries [64]. Reductions in the abundance and richness of carabids were partly attributed to natural evolution [65], which cannot be clearly separated from time-related effects, e.g., structural disturbances [66]. The observed loss may also be due to decreased plant diversity at the terrestrial landscape level or due to habitat loss and vegetation. Alternatively, grassland desertification might be further worsened, caused by climate change, which, as mentioned earlier, had a negative impact on carabid resources.

Environmental data alone are generally deficient in explaining changes in insect communities [67]. We overcame this problem by linking environmental data to the carabid beetle survey data and to the data describing vegetation changes [68]. Making the relationship between environmental and vegetation variables, and their effect on carabid beetles explicit requires careful studies and long-term monitoring. The model revealed that precipitation had significant effects on species abundance and richness in our dataset. On the regional scale, abundance was significantly influenced by precipitation, soil temperature, plant diversity, and latitude, and the species richness was mainly affected by precipitation, soil moisture, and latitude. Moisture changes also affected vegetation, which could have knock-on influences on the micro-climate. The importance of precipitation changes could be explained by the free-ranging lifestyle of immature larval stages. It is known that an increase in precipitation can enhance the aboveground vegetation biomass [69]. Vegetation provides food for herbivorous beetle species and shelters for predatory species and may facilitate the richness of greater carabid species. We found that soil temperature also had a significant effect on abundance because some beetles lay eggs in burrows and others overwinter as larvae or adults in the soil, so a warmer temperature can stimulate the number of beetles, their range of activity, and therefore, beetle abundance [70,71].

In addition to soil temperature and precipitation, altitude also positively affects the variation in beetle abundance. Latitude is correlated with variations in temperature, humidity, and precipitation, so it is a measure of environmental heterogeneity [72]. Carabid beetles show a closer correlation with the latitudinal gradient than other insects. A linear reduction or hump-shaped distribution of arthropod species richness along an altitudinal gradient has been reported previously [73]. Studies have suggested that carabid beetles in Europe have moved tens of meters in elevation in the past 10 to 20 years [74]. In our study, we found that altitude was an important variable in explaining beetle richness. The increase in richness with increasing elevation agrees with previous results because more suitable environmental conditions with appropriate temperature and precipitation ranges for the beetles of northern China are found at higher altitudes. Beetle species richness tends to increase with altitude but not with abundance [75]. Therefore, beetles have a complicated relationship with altitude at the species level, which needs further research. The significance of altitude in this study may indicate an important covariation with temperature and precipitation and may provide a cue for key biological events as temperature and precipitation control the reproduction of carabid beetles [25].

Declines in the carabid species numbers in both the Yanchi and Guyuan regions may lead to the homogenization of beetle communities (Figure 5), which agrees with that of other studies of insects [76]. Our results showed that nutrient-poor grasslands, where many species have declined, should be protected (e.g., the Yanchi region) [77]. Environmental changes and environmental function traits are important explanations for carabid species decline [49,78]. In addition, measures to manage carabid beetles should include protection of grasslands and appropriate nature management. The strict management of habitats can maintain the population stability of carabid species.

Species declines in insects have been reported recently in many countries [7,78]. The decline in carabid beetles was previously strongly related to climate change [72]. Carabid beetles are closely associated with broad habitat types. Species abundance and richness decreased in two regions due to habitat loss and climate change, which was consistent with previous studies [79,80,81]. After 12 years, GY has a broad distribution of carabid species, which is more consistent with our conclusions from previous investigations. Our previous conclusion was that GY undergoes desertification towards becoming a desert steppe with low species richness and weak functional diversity. Therefore, conservation efforts should generally include establishing new habitats and restoring natural vegetation, which needs long-term evolution. An appropriate habitat is crucial for the successful recolonization of beetles [82].

## 5. Conclusions

In conclusion, our findings show that the carabid beetle assemblage of YC suffers serious threats and declines in species and abundance caused by precipitation and soil temperature. The risk of extinction increasingly forces plant and animal species to move to more suitable habitats, which can lead to a significantly different distribution pattern of insect species in the future. It is unclear whether the decline might apply to other groups and regions; there is a need for long-term datasets to be gathered at a global scale, especially in grassland systems. Furthermore, our study is a first attempt at understanding the main drivers of the spatial pattern of insect species richness in temperate grasslands of northwestern China. The methods used here can assess other taxa and can assist managers in planning where conservation efforts need to be focused.

## Figures and Tables

**Figure 1 insects-13-00035-f001:**
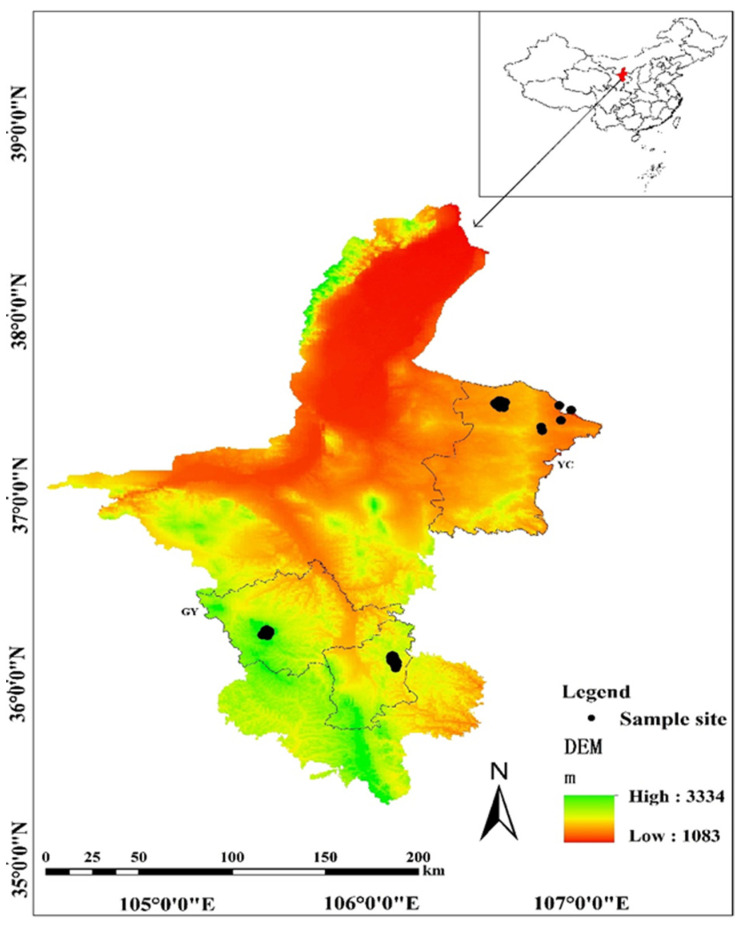
The location of sampling sites and region boundaries within Ningxia Province, northwestern China (YC, Yanchi region; GY, Guyuan region).

**Figure 2 insects-13-00035-f002:**
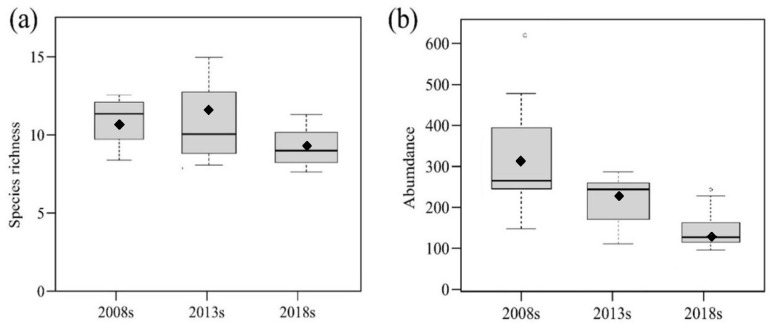
Boxplots of carabid beetle species richness (**a**) and abundance (**b**) per survey at five-year intervals. The upper and lower edges represent upper and lower quartiles, the bold black lines represent median values, and the black rhombuses indicate means.

**Figure 3 insects-13-00035-f003:**
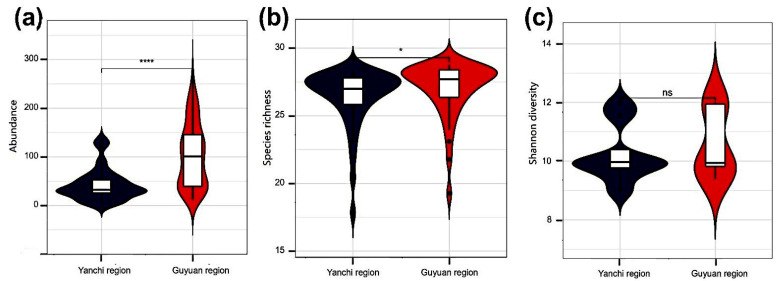
Violin plots showing probability densities of (**a**) abundance, (**b**) species richness, and (**c**) Shannon diversity. The boxplots inside violins represent the distribution of measured data and show the median, and the lower and upper limits. Adjusted *p*-values are indicated as * *p* < 0.05; **** *p* < 0.0001; ns, not significant (*p* > 0.05).

**Figure 4 insects-13-00035-f004:**
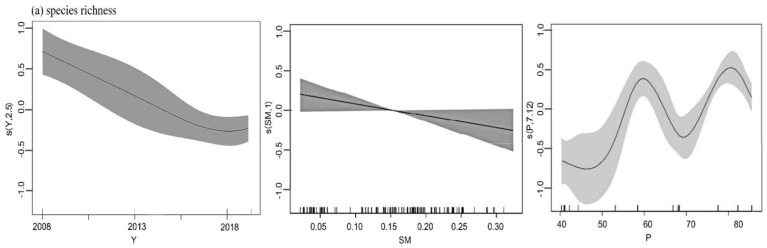
Plots of significant explanatory variables for the dependent variables species richness (**a**) and abundance (**b**). P, precipitation; Lat, latitude; ST, soil temperature; PD, plant diversity; SM, soil temperature. The vertical axes are expressed in logits, the value (s) represents the smoothing fitting value of an explanatory variable of carabid beetles, and the ordinate in parentheses is the estimated degree of freedom. The shaded areas represent confidence intervals.

**Figure 5 insects-13-00035-f005:**
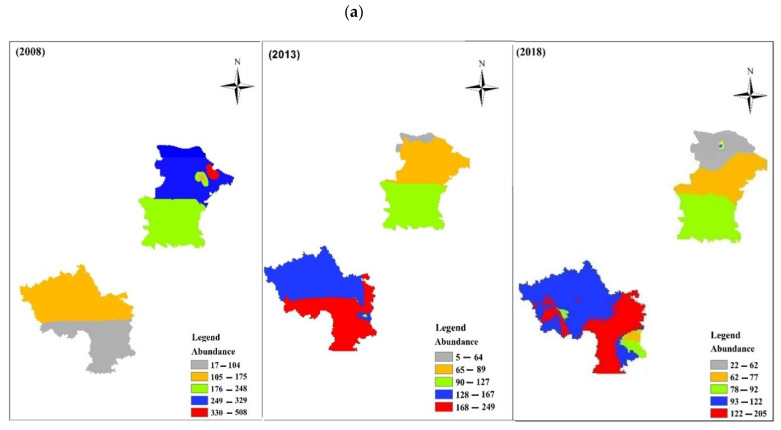
Map of carabid beetle study areas showing (**a**) abundance and (**b**) species richness per survey at five-year intervals.

**Figure 6 insects-13-00035-f006:**
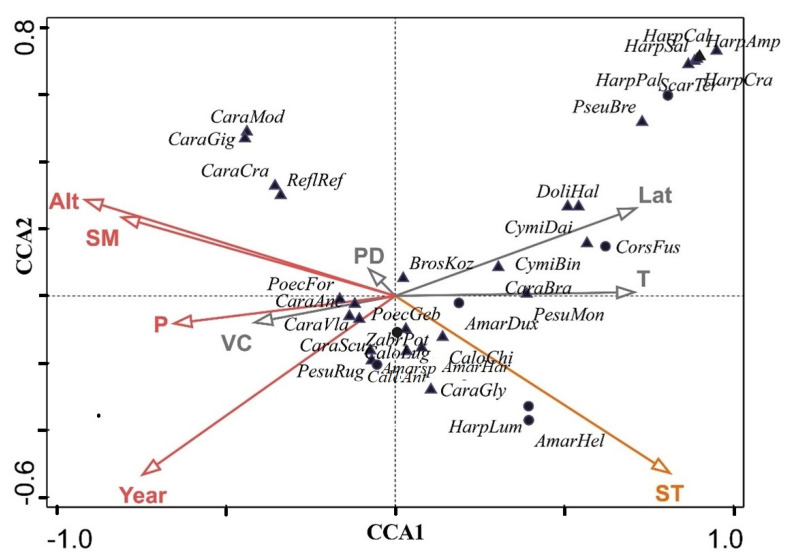
CCA ordination with main gradients marked in red (● = herbivores and ▲ = predators, species abbreviations as in Table A2).

**Table 1 insects-13-00035-t001:** The average species richness and abundance of carabid beetle in Yanchi and Guyuan region (mean number/sample plot, *n* presents sampling plot).

Value	Study Area			
	Yanchi Region (*n* = 31)		Guyuan Region (*n* = 104)	
	Species Richness Individuals/Plot	Abundance	Species Richness	Abundance
Average value	5.26	62.91	9.32	92.95
Standard deviation	3.56	103.85	3.03	64.04
Maximum	17	523	16	268
Minimum	1	2	6	30

**Table 2 insects-13-00035-t002:** Summary of generalized additive regression for the dependent variables of abundance and species richness. *** *p* < 0.001; * *p* < 0.05.

Term	Abbreviation	Species Richness			Abundance		
		Df	F	*p*	Df	F	*p*
Year	Year	2.751	9.871	***	2.760	21.74	***
Latitude	Lat	-	-	-	1.658	5.338	*
Mean annual temperature	T	-	-	-	-	-	-
Mean annual precipitation	P	7.12	8.729	***	5.152	19.516	***
Altitude	Alt	-	-	-	-	-	-
Vegetation coverage	VC	-	-	-	-	-	-
Plant diversity	PD	-	-	-	1.000	6.114	*
Soil moisture	SM	1.00	3.458	*	-	-	-
Soil temperature	ST	-	-	-	2.269	7.264	***
Site	S	-	-	-	-	-	-

Dashes indicate variables that did not contribute to the models and were thus excluded.

## Data Availability

The data presented in this study are available on request from the correspondence author.

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
