# Peer review of "Decrease in Carabid Beetles in Grasslands of Northwestern China: Further Evidence of Insect Biodiversity Loss"

_insects, 2021, doi:10.3390/insects13010035_

Round 1

Reviewer 1 Report

First of all, I applaud the authors for studying insect communities over a ten-year period and for focusing on a hyper-diverse group such as carabid beetles. I was a bit surprised that they only found 34 beetle species with this substantial sampling effort – but this is likely due to the nature of these particular habitat types.

The authors advertise their study as “The decrease of carabid beetles […] further evidence of insect biodiversity loss” and present declines in abundance and species richness of only 0.2% and 1.2% during ten years. This is a very low effect size! Would you consider a 0.2% decline among 35 species as a policy maker? In my experience, we are looking at random variation here. I suspect that significant differences/changes are due to pseudo-replication (possibly due to spatial dependence of single pitfall traps at sites as well as repeated measurements of sites that were not adequately addressed in the statistical analyses) resulting in illegitimate high sampling sizes and consequently low p-values.

The inconsistent vocabulary when addressing sites (sites are used at different scales – sometimes referring to pitfall traps e.g. L 113/Table S1 is about Quadrat/surveys) makes it impossible to comprehend the messy design (messy is ok, as long as it is treated correctly in the analysis). What is the minimum distance between sites? The statistic section provides no information about nested models/random effects and degrees of freedom are not provided!? The authors need to convince readers that these results are indeed significant and not just artefacts by providing more clear information about pooling of data and the number of independent statistical units (sample size n) that entered the respective models!

The authors summarise their findings as “substantial declines in carabid beetles” (L 215, remember my comments above), however, Fig. 3a depicts a unimodal relationship – so if these relationships are indeed significant, it is more complicated, requires a more nuanced discussion, and a more modest framing of the study (contrasting changes in beetle assemblages in different regions…).

I do not understand the separation in three groups depicted in Fig 2 - what is the basis of this combination of surveyed years? Any pair-wise comparisons?

The authors emphasise the importance of multivariate analyses (L 74), however, multivariate results such as homogenization of beetle communities remain speculative (L 285) and are not formally addressed (i.e. by comparing multivariate dispersion/b-diversity of communities). Furthermore, it would be very interesting to see changes in beetle communities along altitudes – were there actually beetle species that moved to higher altitudes during the ten-year period?

In general, the introduction can be shortened by removing repetition (biodiversity~ecosystem services; insects~ecosystem services; indicator species, see detailed comments below) and many parts need language editing (see below).

Detailed comments:

L 9: “Beetles are key insect species” in which sense/what is meant by ‘key insect’? Please be more specific

L 9: “poorly fertile” change to nutrient-poor or nutrient-deficient

L 18-19: “because of less precipitation and loss of habitat including precipitation and habitats.” Unclear to me – please reformulate

L 33: “According to statistics,” omit this part – can be taken for grated in a serious scientific context.

L 39: “Infact, biologists speculate…” awkward/contradictory word combination – please reformulate

L 40-41: “Declines insect abundance declined…” please reformulate

L 42: “most mammals have declined over 80%” this is too general/vague – how many are most (see also L 44)?

L 44: “from a 2019 the assessment” formulation

L 46: “of of” one of is enough

L 54: “To our knowledge, it is a pity that” please formulate more objectively – your feelings are irrelevant here.

L 57-58: The concept of ‘indicator species’ is largely discredited by current meta-analysis: cross-taxa congruence is generally low and random species are equally efficient to e.g. protect other species. Different species respond to different biotic or abiotic factors and are affected at different spatio-temporal scales. Anyway, the topic of indicator species is not relevant for this study – can be omitted.

L 66-67: “in the agricultural ecosystem, it provides important ecosystem services…” what is ‘it’ – please reformulate

L 73-74: “To make predictions about which taxa will be most threatened and to offer information for policy-making, it is essential to utilize multivariate analysis of data.” Makes no sense to me: it requires multi-species comparisons but a univariate approach e.g. population viability analyses of singe species populations would be much more insightful.

L 78-79: omit the bumblebee example – is more a distraction at this point of the ms

L 94: change “probability” to “opportunity”

L 106-108: What is the rational behind hypothesis iii? Sound like a posteriori hypothesis – the interplay between habitat/landscape and climate change/species decline needs more attention in the introduction.

At this point I stopped commenting on the language. Please prepare your manuscript more carefully. The manuscript would greatly benefit from editing by a native speaker.

Author Response

Response to Reviewer 1 Comments

First of all, I applaud the authors for studying insect communities over a ten-year period and for focusing on a hyper-diverse group such as carabid beetles. I was a bit surprised that they only found 34 beetle species with this substantial sampling effort – but this is likely due to the nature of these particular habitat types.

Response1: Thank you. 

The authors advertise their study as “The decrease of carabid beetles […] further evidence of insect biodiversity loss” and present declines in abundance and species richness of only 0.2% and 1.2% during ten years. This is a very low effect size! Would you consider a 0.2% decline among 35 species as a policy maker? In my experience, we are looking at random variation here. I suspect that significant differences/changes are due to pseudo-replication (possibly due to spatial dependence of single pitfall traps at sites as well as repeated measurements of sites that were not adequately addressed in the statistical analyses) resulting in illegitimate high sampling sizes and consequently low p-values.

Response1: Thanks for your suggestion. We thought we made a mistake about the decrease of species richness and abundance. The species richness and abundance per survey were decreased over time by approximately 0.2% and 1.02%, respectively. We feel sorry that this manuscript made you uncertain, and we have modified it in the revision.  

The inconsistent vocabulary when addressing sites (sites are used at different scales – sometimes referring to pitfall traps e.g. L 113/Table S1 is about Quadrat/surveys) makes it impossible to comprehend the messy design (messy is ok, as long as it is treated correctly in the analysis). What is the minimum distance between sites? The statistic section provides no information about nested models/random effects and degrees of freedom are not provided!? The authors need to convince readers that these results are indeed significant and not just artefacts by providing more clear information about pooling of data and the number of independent statistical units (sample size n) that entered the respective models!

Response1: Thank you. We are sorry to make those mistakes. In each site, five pitfall traps were monitored on five dates from May to September. Each sample site consisted of five pitfall traps at 5-m intervals, which separated by at least 200 m from each other to minimize spatial autocorrelation. All sites and years of the number of sampling rounds and their dates were synchronized. Yes, thanks for your constructive suggestions. We have supplemented it in the revision.

The authors summarise their findings as “substantial declines in carabid beetles” (L 215, remember my comments above), however, Fig. 3a depicts a unimodal relationship – so if these relationships are indeed significant, it is more complicated, requires a more nuanced discussion, and a more modest framing of the study (contrasting changes in beetle assemblages in different regions…).

Response1: Thanks for your constructive suggestions. We have supplemented the abundance, species richness, and diversity in the different regions in the revision. The beetle assemblages in different regions can be seen in Table S2. 

I do not understand the separation in three groups depicted in Fig 2 - what is the basis of this combination of surveyed years? Any pair-wise comparisons?

Response1: Thank you. We surveyed the data from 2008, 2009, 2013, 2017, to 2019, so we drew a map per five years.  

The authors emphasise the importance of multivariate analyses (L 74), however, multivariate results such as homogenization of beetle communities remain speculative (L 285) and are not formally addressed (i.e. by comparing multivariate dispersion/b-diversity of communities).

Furthermore, it would be very interesting to see changes in beetle communities along altitudes – were there actually beetle species that moved to higher altitudes during the ten-year period?

Response1: Thank you. We accepted your suggestion to pay attention to beetle species that moved to higher altitudes. Of course, we have made a corresponding modification in the discussion.

In general, the introduction can be shortened by removing repetition (biodiversity~ecosystem services; insects~ecosystem services; indicator species, see detailed comments below) and many parts need language editing (see below).

Response1: Thank you. We have modified by language company.

L 9: “Beetles are key insect species” in which sense/what is meant by ‘key insect’? Please be more specific

Reply: Thank you. Yes, we thought that carabid beetles is a key species which can indicate the change of steppe, so we used a key insect species to show. We thought it was error, so we have modified in the revision.

L 9: “poorly fertile” change to nutrient-poor or nutrient-deficient

Reply: Thank you. We have modified in the revision.

L 18-19: “because of less precipitation and loss of habitat including precipitation and habitats.” Unclear to me – please reformulate

Reply: Thank you. We have modified in the revision.

L 33: “According to statistics,” omit this part – can be taken for grated in a serious scientific context.

Reply: Thanks for your suggestion. We have modified in the revision.

L 39: “Infact, biologists speculate…” awkward/contradictory word combination – please reformulate

Reply: Thank you. We have modified in the revision.

L 40-41: “Declines insect abundance declined…” please reformulate

Reply: Thanks for your suggestion. We have modified in the revision.

L 42: “most mammals have declined over 80%” this is too general/vague – how many are most (see also L 44)?

Reply: Thank you. We have modified in the revision.

L 44: “from a 2019 the assessment” formulation

Reply: Thank you. We have modified in the revision.

L 46: “of of” one of is enough

Reply: Thank you. We have modified in the revision

L 54: “To our knowledge, it is a pity that” please formulate more objectively – your feelings are irrelevant here.

Reply: Thanks for your suggestion. We have modified in the revision.

L 57-58: The concept of ‘indicator species’ is largely discredited by current meta-analysis: cross-taxa congruence is generally low and random species are equally efficient to e.g. protect other species. Different species respond to different biotic or abiotic factors and are affected at different spatio-temporal scales. Anyway, the topic of indicator species is not relevant for this study – can be omitted.

Reply: Thanks for your suggestion. We have modified in the revision.

L 66-67: “in the agricultural ecosystem, it provides important ecosystem services…” what is ‘it’ – please reformulate

Reply: Thank you. We have modified in the revision.

L 73-74: “To make predictions about which taxa will be most threatened and to offer information for policy-making, it is essential to utilize multivariate analysis of data.” Makes no sense to me: it requires multi-species comparisons but a univariate approach e.g. population viability analyses of singe species populations would be much more insightful.

Reply: Thanks for. We want to show that is  essential  to use multivariate analysis, we thought it is insignificance in this sentence,  we have modified in the revision.

L 78-79: omit the bumblebee example – is more a distraction at this point of the ms

Reply: Thank you. We have modified in the revision.

L 94: change “probability” to “opportunity”

Reply: Thank you. We have modified in the revision.

L 106-108: What is the rational behind hypothesis iii? Sound like a posteriori hypothesis – the interplay between habitat/landscape and climate change/species decline needs more attention in the introduction.

Reply: Thank you. the few existing studies use different measures of biodiversity (e.g. biomass, abundance or species numbers), it is difficult to derive reliable conservation strategies for ground beetles in woodland habitats. Thus, our study contributes to a better understanding of long-term dynamics of ground beetle communities for our region.

Reviewer 2 Report

Dear Authors,

I sincerely appreciate the effort in collecting and analyzing all those data, and I am sure that the content of the paper will be of primary interest to the scientific community.

However, in its present state, the draft requires further work both on style  (phrasing, mistakes, etc..) and scientific content. In the specific regard of this last topic, I would like the authors could better explain the results in the discussion, linking model results to carabid biology in a more exhaustive manner, and if necessary to discuss in detail some of the species they collected. 

IMPORTANT: the simple summary is mandatory in INSECTS, following guidelines your draft should have neem rejected

The references are not consistent with MDPI guidelines

Author Response

Response to Reviewer 2 Comments

However, in its present state, the draft requires further work both on style  (phrasing, mistakes, etc..) and scientific content.  In the specific regard of this last topic, I would like the authors could better explain the results in the discussion, linking model results to carabid biology in a more exhaustive manner, and if necessary to discuss in detail some of the species they collected.

Reply: Thanks for your suggestion. We have modified in the revision.

IMPORTANT: the simple summary is mandatory in INSECTS, following guidelines your draft should have neem rejected

 Reply: Thank you. We have modified in the revision.

The references are not consistent with MDPI guidelines

Reply: Thank you. We have modified in the revision.

Round 2

Reviewer 1 Report

I thank the authors for improving their manuscript and for responding to my questions/suggestions. The terminology is much clearer now, which helps to understand the study design. There was a lot of apologizing for what the authors called “mistakes” in the author’s response. However, this does not help to improve the contribution. The authors need to transparently show relevant information in the manuscript (which is already improved), the statistical analysis needs to be correct and weaknesses/conflicting evidence needs to be discussed adequately. I do not want to sound mean but readers need to be able to trust the study and to evaluate its quality.

My main point of criticism is still pseudo-replication in space and in time (see the classic paper by Hurlbert 1984, Ecological Monographs 54:187-211) and the sample size that actually entered the models (Table 1 is clear for me [n represents site] but I am interested in the n/df of Table 2). On statistical grounds, it is NOT meaningful to analyze the data per trap because these are not independent samples (even 200 m between the sites is pretty close for carabids…). The easiest way to deal with this is to pool the traps per site. Otherwise the models must be nested per site (i.e. to include site identity as a random variable in a GLMM to account for spatial dependence). This must be done anyway because the authors include their sites several times in the different years (i.e. repeated measurements). As it is, the authors treat their samples as if they would be independent and that is simply not the case. Hence, low p-values despite marginal effect size.

If the change among the twelve years remains significant in an adequate model (that incorporates spatial and temporal dependencies) -  which I doubt, then the authors should discuss that a decline of approx. 2 % in twelve years is lower compared to the insect declines reported in other studies (e.g. Hallmann et al. 2017, PLOS ONE doi.org/10.1371/journal.pone.0185809; Seibold et al. 2019, Nature 574:671-674 and others cited in the manuscript) and biodiversity conservation practitioner might focus their effort to reduce insect decline on regions where the decline is much steeper (e.g. Central European farmland).

I think it is save to formulate hypothesis (i) as to expect a decline and not as a question. (L 94)

Fig. 4: What is “Effect” and what is indicated by the grey area? More information in the figure caption needed. Would be nice to see the actual data points.

Author Response

Response to Reviewer 1 Comments

I thank the authors for improving their manuscript and for responding to my questions/suggestions. The terminology is much clearer now, which helps to understand the study design. There was a lot of apologizing for what the authors called “mistakes” in the author’s response. However, this does not help to improve the contribution. The authors need to transparently show relevant information in the manuscript (which is already improved), the statistical analysis needs to be correct and weaknesses/conflicting evidence needs to be discussed adequately. I do not want to sound mean but readers need to be able to trust the study and to evaluate its quality.

Reply: Thank you.

My main point of criticism is still pseudo-replication in space and in time (see the classic paper by Hurlbert 1984, Ecological Monographs 54:187-211) and the sample size that actually entered the models (Table 1 is clear for me [n represents site] but I am interested in the n/df of Table 2). On statistical grounds, it is NOT meaningful to analyze the data per trap because these are not independent samples (even 200 m between the sites is pretty close for carabids…). The easiest way to deal with this is to pool the traps per site. Otherwise, the models must be nested per site (i.e. to include site identity as a random variable in a GLMM to account for spatial dependence). This must be done anyway because the authors include their sites several times in the different years (i.e. repeated measurements). As it is, the authors treat their samples as if they would be independent and that is simply not the case. Hence, low p-values despite marginal effect size.

Reply: Thanks for your suggestion. we understood it. Yes, we have pooled the sample site and received 79 effective sample sites in order to avoid the spatial autocorrelation. Of course, the activity of carabid beetle ranges from 10 meters. We pooled the data per trap in 20×20 m, We analyzed the data again and modified the table 2, fig.2.

If the change among the twelve years remains significant in an adequate model (that incorporates spatial and temporal dependencies) - which I doubt, then the authors should discuss that a decline of approx. 2 % in twelve years is lower compared to the insect declines reported in other studies (e.g. Hallmann et al. 2017, PLOS ONE. doi.org/10.1371/journal.pone.0185809; Seibold et al. 2019, Nature 574:671-674 and others cited in the manuscript) and biodiversity conservation practitioner might focus their effort to reduce insect decline on regions where the decline is much steeper (e.g. Central European farmland).

Reply: Thanks for your suggestion. yes, Concerns are exacerbated for carabid beetle because they play an important role within ecosystem. Although the decline was lower than other insects and the re-percussion for the functional of ecosystem was unknown, the decline of carabid beetles raise the concerns for crop because they are natural enemies of pests and their services are important to agriculture.

I think it is save to formulate hypothesis (i) as to expect a decline and not as a question. (L 94)

Reply: Thank you. We have modified it in the revised version.

Fig. 4: What is “Effect” and what is indicated by the grey area? More information in the figure caption needed. Would be nice to see the actual data points.

Reply: Thank you. We have modified it in the revised version.

Reviewer 2 Report

Dear Authors,

despite important addition to the text, there are still a couple of sentences to be revised (not clear what you mean), plus the daft attched is still missin the short summary (that is different from the abstract) and the references are not yet fully complaint to MDPI guidelines

Author Response

Response to Reviewer 2 Comments

despite important addition to the text, there are still a couple of sentences to be revised (not clear what you mean), plus the daft attched is still miss in the short summary (that is different from the abstract) and the references are not yet fully complaint to MDPI guidelines

Reply: Thank you. We have modified in the revision.

IMPORTANT: the simple summary is mandatory in INSECTS, following guidelines your draft should have neem rejected

 Reply: Thank you. We have modified in the revision.

The references are not consistent with MDPI guidelines

Reply: Thank you. We have modified in the revision.

Round 3

Reviewer 1 Report

L 23-24: Are these changes per year or among the whole duration of 12 years? Please make this clear in the abstract. The percentage of abundance change remained the same after re-analyzes?! Are this numbers correct (see also L 197)?

L 112 vs. L 147: It seems that the analysis have improved but why is there a “effective number of sampling sites”? Why not just the 135 sites - after all they are separated by at least 200 m (L142). I am sorry about this but it is just really confusing…

L 168: As I suggested in previous comments, the analysis should be improved by including site identity as a random effect because some sites are repeatedly measured. The effective sample size of the models also needs to be 135 (79). As it is, the model is based on a sample size that is the number of samples, which are not all independent from each other and should be nested in sites to avoid pseudo-replication (low p-values despite marginal effect size). This might even help to get stronger results because it accounts for variation between the sites.

Fig. 4: The y-axes annotations remain cryptic to me. What is this? Please add meaningful y-axes annotations.

L 248-249: Contradictory formulation “In contrast…, which were in accordance with” please reformulate.

L 252-254: “the repercussion for the functional of ecosystem was unknown […], the decline of carabid beetles raise the concerns for food because…” it is the functioning of ecosystems, but still remains unclear to me. Carabids are one group of generalist predators among others – the relationship between predator abundance/species richness and food production cannot be taken for granted and is often context dependent (see Griffin et al. Ecology, 94(10), 2013, pp. 2180–2187 for a meta-analysis on the topic). Please formulate more carefully.

L 353: “caused by climate changes” too strong – an experimental approach would be required to show causal relevance. “climate change” is too general and can mean many things. Please be more specific and focus on the variables that are actually quantified in the study.

L 356-357: “regular sampling and monitoring in management plans will correctly assess” awkward formulation – regular sampling is needed to monitor species distributions? Please reformulate.

Thanks for your effort!

Author Response

L 23-24: Are these changes per year or among the whole duration of 12 years? Please make this clear in the abstract. The percentage of abundance change remained the same after re-analyzes?! Are this numbers correct (see also L 197)?

Reply: Thank you. We have modified it in the revised version.

L 112 vs. L 147: It seems that the analysis have improved but why is there a “effective number of sampling sites”? Why not just the 135 sites - after all they are separated by at least 200 m (L142). I am sorry about this but it is just really confusing…

Reply: Thank you. Yes, there were 135 sites in the two regions and each sample site was separated by 200–1400 m. In previous manuscript, we analyzed by each sites, in previous comments, “(even 200 m between the sites is pretty close for carabids…).”,therefore, we pooled data the sample sites only interval 200m, we received 79 effective sampling site.

L 168: As I suggested in previous comments, the analysis should be improved by including site identity as a random effect because some sites are repeatedly measured. The effective sample size of the models also needs to be 135 (79). As it is, the model is based on a sample size that is the number of samples, which are not all independent from each other and should be nested in sites to avoid pseudo-replication (low p-values despite marginal effect size). This might even help to get stronger results because it accounts for variation between the sites.

Reply: Thank you. We set the plot as a random effect in the model. But we also see the same result, so we analyzed it again and modified it in the revised version.

Fig. 4: The y-axes annotations remain cryptic to me. What is this? Please add meaningful y-axes annotations.

Reply: Thank you. The vertical axes are expressed in logits and the value (s) represents the smoothing fitting value of explanatory variable of carabid beetle, the ordinate in parentheses is the estimated degrees of freedom.

L 248-249: Contradictory formulation “In contrast…, which were in accordance with” please reformulate.

Reply: Thank you. We have modified it in the revised version.

L 252-254: “the repercussion for the functional of ecosystem was unknown […], the decline of carabid beetles raise the concerns for food because…” it is the functioning of ecosystems, but still remains unclear to me. Carabids are one group of generalist predators among others – the relationship between predator abundance/species richness and food production cannot be taken for granted and is often context dependent (see Griffin et al. Ecology, 94(10), 2013, pp. 2180–2187 for a meta-analysis on the topic). Please formulate more carefully.

Reply: Thanks for your good suggestion. We have modified it in the revised version

L 353: “caused by climate changes” too strong – an experimental approach would be required to show causal relevance. “climate change” is too general and can mean many things. Please be more specific and focus on the variables that are actually quantified in the study.

Reply: Thank you. We have modified it in the revised version.

L 356-357: “regular sampling and monitoring in management plans will correctly assess” awkward formulation – regular sampling is needed to monitor species distributions? Please reformulate.

Reply: Thank you. We have modified it in the revised version.
